# TextSquare: Scaling up Text-Centric Visual Instruction Tuning

## Abstract

Text-centric visual question answering (VQA) has made great strides with the development of Multimodal Large Language Models (MLLMs), yet open-source models still fall short of leading models like GPT4V and Gemini. A key contributing factor to this disparity is the absence of extensive, high-quality instruction tuning data. To this end, we introduce a new approach for creating a massive, high-quality instruction-tuning dataset, Square-10M, generated by leveraging the versatile multimodal capabilities of closed-source MLLMs. The data construction process, termed Square, consists of four steps: **S**elf-**Qu**estioning, **A**nswering, **R**easoning, and **E**valuation. Our experiments with Square-10M led to three key findings: 1) Our model, TextSquare, considerably surpasses open-source previous state-of-the-art text-centric MLLMs and sets a new standard on OCRBench (62.2%). It even outperforms top-tier models like GPT4V and Gemini on six out of ten text-centric benchmarks. 2) We demonstrate the importance of VQA reasoning data in offering comprehensive contextual insights for specific questions, improving accuracy and substantially mitigating hallucinations. Specifically, TextSquare scores an average of 75.1% across four general VQA and hallucination evaluation datasets, outperforming previous state-of-the-art models. 3) Notably, the phenomenon observed in scaling text-centric VQA datasets reveals a vivid pattern: an exponential increase of instruction tuning data volume is directly proportional to the improvement in model performance, thereby validating the necessity of the dataset scale and the high quality of Square-10M.

## 1 Introduction

Recent research on multimodal large language models (MLLMs) (Ye et al., 2023a; Feng et al., 2023b; Liu et al., 2024d; Feng et al., 2023a) has yielded significant advancements in text-centric visual question-answering(VQA), with several closed-source state-of-the-art (SOTA) models (OpenAI, 2023; DeepMind, 2023) leading the way. Two representative examples are GPT4V (OpenAI, 2023) and Gemini (DeepMind, 2023), which have shown exceptional performance and even surpassed human capabilities in some aspects. Nevertheless, as illustrated in Figure 1, open-source models still significantly trail behind their closed-source counterparts. This gap can be attributed to various factors, including model architecture, the scale of model parameters, image resolution, the volume of pretraining and instruction tuning data, and training strategies.

Recent studies (Chen et al., 2024; Nayak et al., 2024; Chen et al., 2023; Zhang et al., 2023) have delved into the challenges of insufficient instruction tuning data. For instance, Monkey (Li et al., 2023c) employed expert models to generate image descriptions, which GPT-4 then summarized to create high-quality, detailed image captions. LLaVAR (Zhang et al., 2023) and TG-Doc (Wang et al., 2023) used GPT-4 to generate conversations for text-rich images by integrating OCR results into the instructions. ShareGPT4V (Chen et al., 2023) constructs a high-quality image caption dataset through GPT4V to improve the image caption ability for MLLMs. While these efforts have achieved remarkable success, they also left some challenges unresolved. Image caption data and VQA data belong to different domains, with inconsistencies in the granularity and scope of image content presentation. Furthermore, the scale of synthetic data remains relatively small, preventing MLLMs from fully realizing their potential. The exploration of methods that leverage large-scale text-centric VQA data for instruction tuning remains limited.

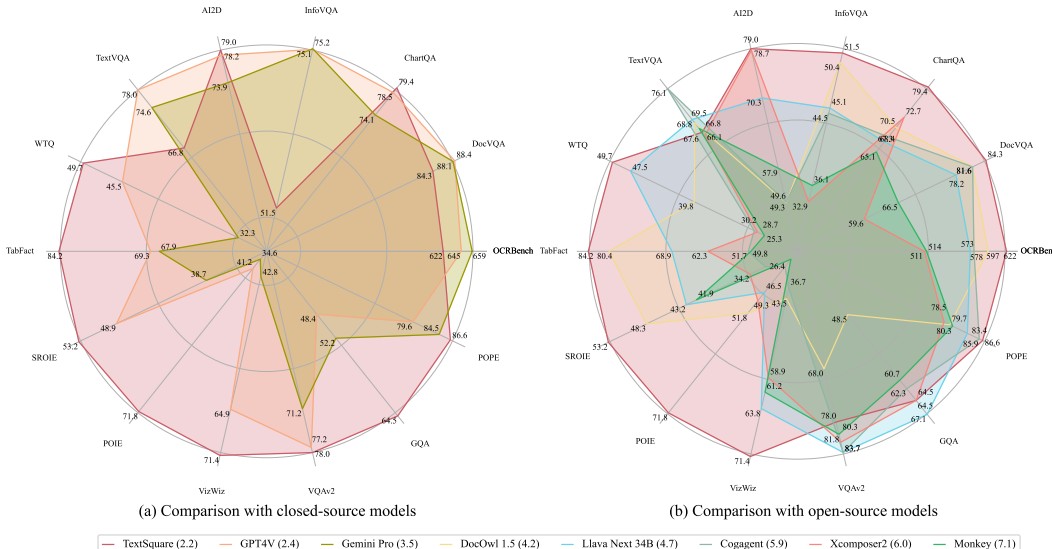

Figure 1: The performance of TextSquare in various VQA tasks compared to existing models. (a) shows the comparison with state-of-the-art closed-source models (Gemini (DeepMind, 2023) and GPT4V (OpenAI, 2023)), and (b) shows the comparison with the leading open-source models. The numbers in parentheses after model names in the legend indicate the average performance ranking across 10 text-centric benchmarks. TextSquare is marginally superior to GPT4V. Best viewed on screen.

To bridge the gap, this paper proposes a strategy termed Square to acquire extensive, high-quality text-centric VQA data from advanced closed-source MLLMs, constructing a dataset (Square-10M) comprising tens of millions of instances for instruction tuning. The Square strategy consists of four steps: Self-Questioning, Answering, Reasoning, and Evaluation. The self-questioning step involves utilizing the MLLM's capabilities in text-image analysis and understanding to generate textual-related questions. The answering step involves answering these questions, leveraging various prompting techniques such as Chain-of-Thought and few-shot prompting. The reasoning step entails probing the model for the reasoning behind its answers, leveraging the powerful reasoning abilities of MLLMs. The evaluation step involves evaluating the question-answer pairs, assessing the validity of the questions, the relevance to the textual content of images, and the correctness of answers, thereby improving data quality and mitigating hallucinations. Overall, Square comprehensively leverages the various capabilities of MLLMs, significantly enhancing data quality.

Besides, enriching the diversity of images is also crucial. We collect a diverse set of text-rich images from various public sources, including natural scenes, charts, tables, receipts, books, slides, PDFs, documents, products, and web images. Subsequently, deduplication is performed on this collection. By applying the Square strategy to these images, Square-10M is constructed.

Based on Square-10M, our model (TextSquare) achieves remarkable results. First, as shown in Figure 1, TextSquare performs comparably or even better than advanced closed-source models and substantially surpasses recent open-source models on various benchmarks. Notably, the image resolution of TextSquare is 700, and the parameters are 8.6B. Second, our experiments validate the beneficial impact of reasoning data on VQA tasks, demonstrating its ability to enhance model performance while mitigating hallucinations. With reasoning data for instruction tuning, TextSquare has a strong reasoning capability to provide elaborate explanations in VQA scenarios. Additionally, our scaling experiments reveal the relationships between instruction tuning data scale, training convergence loss, and model performance. Whereas a few instruction tuning data can effectively engage MLLMs, it is insufficient. Large amounts of high-quality data can further significantly reduce convergence loss and improve performance. The performance of TextSquare grows and the loss of convergence decreases while continuously scaling up the instruction tuning data, which also demonstrates the effectiveness of Square-10M.

In summary, the main contributions of this paper can be categorized into four points:

- A high-quality dataset (Square-10M) comprising tens of millions of instances for text-centric VQA instruction tuning is constructed by collecting diverse text-rich images and employing the Square (Self-Questioning, Answering, Reasoning, and Evaluation) strategy on closed-source MLLMs.

- Leveraging Square-10M, TextSquare achieves a significant outperformance of existing open-source models and even comparable or superior performance to SOTA closed-source models on various benchmarks, e.g., +0.9% on ChartQA, +2.1% on WTQ, +4.3% on SROIE. TextSquare outperforms GPT4V in overall rankings across ten text-centric benchmarks (ranking 2.2 *v.s.* 2.4).

- Reasoning data is demonstrated to be beneficial in improving model performance and mitigating hallucinations in VQA scenarios, as it can deliver rich question-specific contextual information.

- Through extensive experiments, we reveal the relationships between data scale, convergence loss, and model performance for text-centric VQA instruction tuning, demonstrating the effectiveness and necessity of Square-10M.

## 2    RELATED WORK

Related work is detailed in Section A.2 of the Supplementary Material.

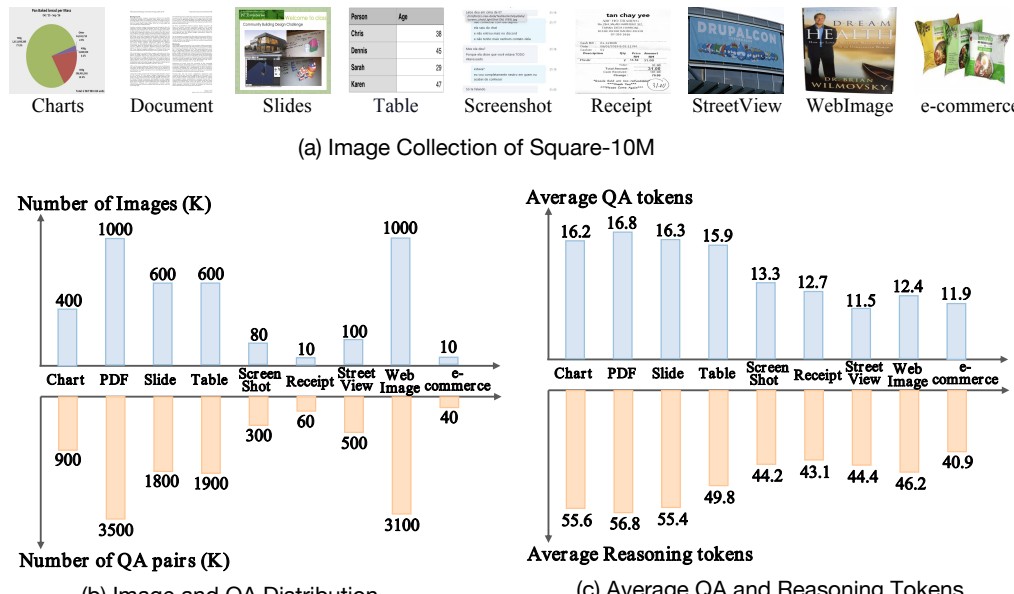

Figure 2: Overview of Square-10M: the distribution of images, the average tokens of the QAs, etc.

## 3    SQUARE-10M: A MASSIVE AND HIGH-QUALITY TEXT-CENTRIC VQA INSTRUCTION TUNING DATASET

Square-10M is synthesized by our proposed Square pipeline, *i.e.*, Self-Questioning, Answering, Reasoning, and Evaluation.

### 3.1    OVERVIEW OF THE SQUARE STRATEGY

Figure 3 presents an overview of our proposed Square. Square generally consists of three stages for synthesizing high-quality instruction tuning data for text-centric VQA: (1) Data Collection: we gather a vast collection of text-rich images. (2) Data Generation: it involves self-questioning, answering, and reasoning, utilizing the images procured. In this phase, the MLLM generates VQA pairs predicated on the images, accompanied by the rationale behind the answers. (3) Data Filtering: we focus on eliminating nonsensical questions and erroneous answers by leveraging the evaluation capabilities of MLLMs.

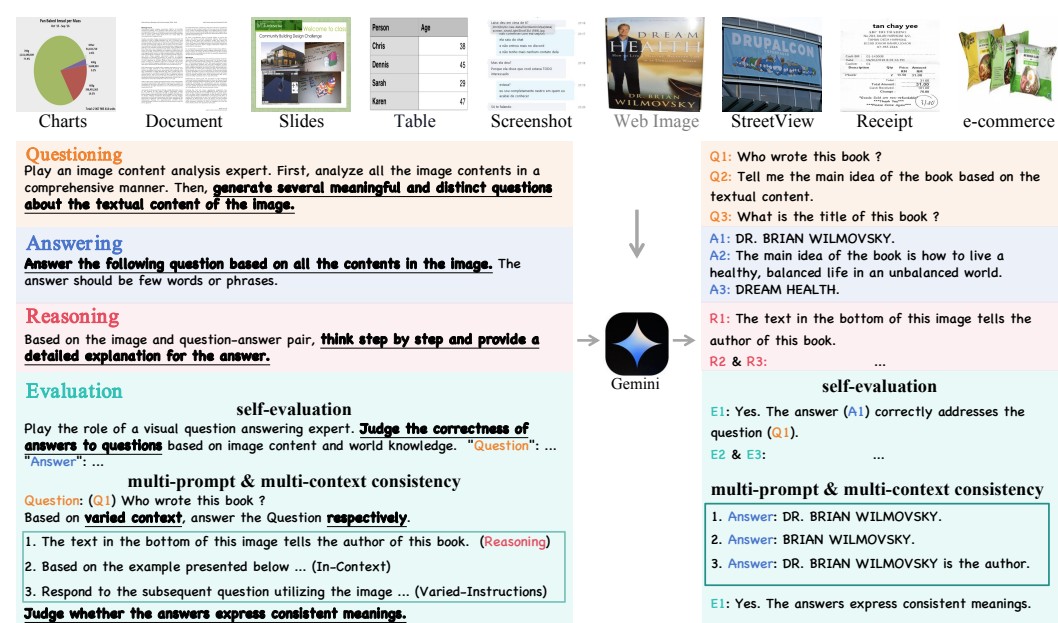

Figure 3: Pipeline of the proposed Square strategy. Gemini's versatile multi-modal capabilities are utilized with prompt engineering to synthesize Square-10M, which consists of four stages: self-questioning, answering, reasoning, and evaluation.

These procedures culminate with the Square-10M dataset, distinguished by its extensive array of high-quality text-centric VQA pairs and associated reasoning context. Specifically, we amass 3.8 million images with varied textual elements from multiple sources. This yields 20 million question-answer pairs during the Data Generation phase. After rigorous filtering, we distill 9.8 million QA pairs along with their reasoning context, employing our Square strategy. The samples filtered out by each strategy are listed below: 4.9 million by Self-Evaluation of MLLMs, 2.1 million by Multi-Prompt Consistency, and 3.2 million by Multi-Context Consistency. The Square-10M dataset is further analyzed in Figure 2.

## 3.2 DATA COLLECTION

The data collection process aims to cover a variety of text-rich scenarios in the real world. We collect 3.8 million unlabeled text-rich images (Figure 2), showcasing diverse properties. For instance, images categorized as Chart and Table focus on textual elements with intense statistical information; Slide, Screenshot, and WebImage are tailored for the interaction between text and prominent visual messages; Document, PDF, Receipt, and e-commerce images are characterized by fine, dense text; Street-View images are derived from natural scenes. This comprehensive image collection provides a representative cross-section of text-rich images in real-world scenarios, forming the foundation of our research on text-centric VQA.

## 3.3 DATA GENERATION: SELF-QUESTIONING, ANSWERING, AND REASONING

To construct the Square-10M dataset, we harness the multi-modal understanding capabilities of Gemini, one of the most advanced LLMs. For each selected image, Gemini generates VQA pairs and the reasoning context through three stages:

**Stage 1: Self-Questioning.** Gemini is prompted to formulate profound, meaningful questions about each image. We prompt Gemini to first comprehensively analyze the image and generate questions based on its interpretation, as shown in Figure 3. To bolster understanding of visual text, we also incorporate text extracted via OCR expert models into the prompts.

**Stage 2: Answering.** Gemini is then directed to answer the generated questions. We leverage various prompting techniques to enrich contextual information and improve the reliability of the responses, such as Chain-of-Thought and few-shot prompting, exemplified in Figure 3.

**Stage 3: Reasoning.** We require Gemini to elucidate the reasoning behind its answers. This process fosters a deeper connection between the questions and visual elements, mitigating hallucinations and ensuring accurate responses. The reasoning also provides additional context for individual questions, potentially aiding research into in-context learning mechanisms. An illustrative prompt for self-reasoning is presented in Figure 3.

### 3.4 DATA FILTERING: SELF-EVALUATION AND ANSWERING CONSISTENCY

Despite the efficacy of self-questioning, answering, and reasoning, some generated content may be hallucinatory or contain meaningless questions and erroneous answers. We establish filtering rules based on the evaluative capabilities of LLMs to select high-quality VQA pairs. This comprehensive filtering system encompasses three aspects:

**Self-Evaluation of MLLMs.** Advanced MLLMs, including Gemini, are utilized to assess the meaningfulness of questions and the adequacy of answers. An example of self-evaluation prompting is illustrated in Figure 3.

**Multi-Prompt Consistency.** We augment the prompt and context space during Data Generation, ensuring that a valid VQA pair remains semantically consistent under diverse prompts. If answers vary significantly in meaning, the VQA pair is discarded, as shown in Figure 3.

**Multi-Context Consistency.** Similar to Multi-Prompt Consistency, VQA pairs are further verified by appending varied contexts to the question. Given the generated question, three types of answers are produced by Gemini with different contexts: (1) Answering with reasoning. Gemini answers the question with a detailed explanation prepended (*i.e.*, content generated in the stage of Reasoning). (2) In-Context answering. Gemini answers the question with chain-of-thought or few-shot prompts prepended. (3) Naive answering. Gemini answers the question with no extra context. VQA pairs will be removed if the generated answers are not semantically consistent.

### 3.5 MLLM SELECTION

Since a comprehensive comparison of all available VLMs to find the optimal VLM is not possible considering the scale of the dataset, we apply the Square strategy to different VLMs (including Gemini-pro, GPT-4V, Qwen-VL-Plus, and Claude 3) and perform a manual comparison on the sampled data. We collect 1,000 QA pairs from each VLM, and perform human evaluation on the generated data. Specifically, for each VLM, the questionnaire consists of 1,000 cases, each of which includes a "Yes or No" question: Is the "Question" meaningful to the image and can the "Answer" correctly respond to the "Question" ? Overall, we have collected 10 questionnaires, and the results are Gemini-pro (94.9%), GPT-4V(95.2%), Qwen-VL-Plus(92.8%),and Claude 3(92.1%). Considering the time cost, price and quality of the data generated, Gemini-pro is our best choice for a full-scale attempt at the Square strategy.

### 3.6 DATA QUALITY EVALUATION: HUMAN VERIFICATION

**Harmful information.** In order to minimize the proportion of harmful information, we have set the Gemini's security level to the maximum. Besides, our dataset is about visual text and with quality assessment and adequate filtering, there is little harmful content in the dataset. Considering the large size of the dataset, we did not have enough resources to conduct a full human assessment. We sampled the dataset (100,000 samples) and did not find harmful information.

**Faulty information.** Faulty information is almost unavoidable for very large generated datasets(e.g., ShareGPT4V[1], Monkey[2]), and there is no guarantee that even manually labelled data is completely correct. To verify the effectiveness of the Square strategy in eliminating factual errors, we performed a manual evaluation on 1,000 samples. Our proposed Square strategy improves the accuracy of the generated data from 82.6% to 94.9%, significantly reducing the probability of faulty information. What's more, TextSquare greatly mitigates the model hallucinations, which is beneficial to the development of MLLMs.

## 4 TEXTSQUARE: A TEXT-CENTRIC MULTIMODAL LARGE LANGUAGE MODEL

### 4.1 MODEL ARCHITECTURE

TextSquare's architecture follows the framework of InternLM-Xcomposer2 (Dong et al., 2024), comprising three integral components: (1) A vision encoder modified from OpenAI CLIP ViT-L (Radford et al., 2021), with an increased resolution of 700 to better capture fine-grained features. (2) A LLM based on InternLM2 (Cai et al., 2024), utilizing InternLM2-7B-Chat as the practical variant. (3) A projector that semantically aligns vision and text.

### 4.2 SUPERVISED FINE-TUNING WITH SQUARE-10M

TextSquare is achieved by performing Supervised Fine-Tuning (SFT) with Square-10M. The SFT process entails three stages: initially, all components are unfrozen and trained at a resolution of 490. Subsequently, we increase the input resolution to 700 and focus on training the Vision Encoder to adapt to the higher resolution. In the final stage, full-parameter fine-tuning is performed at a resolution of 700. TextSquare demonstrates that with our Square-10M dataset, a model with 8B parameters and normal-size image resolution can perform exceptionally on text-centric VQA tasks, outperforming most available MLLMs and even closed-source SOTA models.

## 5 EXPERIMENT

### 5.1 IMPLEMENTATION DETAILS

The training data contains Square-10M and in-domain datasets (consistent with Monkey's SFT data). The training process is divided into three phases, using the same data and the AdamW (Loshchilov & Hutter, 2017) optimizer with 64 A100-80G GPUs. In the first phase, we fine-tune InternLM-Xcomposer2 with full parameters, and the learning rate decreases from 1e-5 to 1e-6, taking about 9520 GPU hours. In the second phase, we scale up the image resolution to 700 and train only VIT, with the learning rate decreasing from 1e-4 to 1e-5, taking about 7280 GPU hours. In the third stage, we perform full-parameter fine-tuning at 700 image resolution, and the learning rate drops from 1e-5 to 1e-6, spending about 12350 GPU hours.

### 5.2 BENCHMARK EVALUATION

We report the results on Scene Text-centric VQA, Document-oriented VQA, Table VQA, Text-centric KIE, OCRBench, and General VQA for a comprehensive comparison of the performance of our model with existing models. The metrics of each benchmark are listed in Table 8 in the Supplementary Material.

Table 1: Quantitative comparison of TextSquare with existing MLLMs on various text-centric benchmarks. "Res." denotes image resolution. "*" denotes the results obtained through the open-source checkpoint or API of the closed-source model. The best results of each benchmark are **bolded**. The best results except for closed-source models (GPT4V and Gemini Pro) are underlined.

| Method | Res. | OCRBench | Document-Oriented | | | Scene Text-Centric | | Table VQA | | KIE | |
| | | | DocVQA | ChartQA | InfoVQA | AI2D | TextVQA | WTQ | TabFact | SROIE | POIE |
|---|---|---|---|---|---|---|---|---|---|---|---|
| UReader (Ye et al., 2023b) | 896 | - | 65.4 | 59.3 | 42.2 | - | - | - | - | - | - |
| Qwen-VL (Bai et al., 2023) | 448 | 506 | 65.1 | 65.7 | - | - | 63.8 | - | - | - | - |
| TextMonkey (Liu et al., 2024d) | 896 | 558 | 73.0 | 67.1 | - | 44.7 | 65.6 | 37.9 | 53.6 | 46.2 | 32.0 |
| Monkey (Li et al., 2023c) | 896 | 514 | 66.5 | 65.1 | 36.1 | 57.9* | 67.6 | 25.3* | 49.8 | 41.9 | 19.9 |
| Cogagent (Hong et al., 2023) | 1120 | 578* | 81.6 | 68.4 | 44.5 | 49.6* | 76.1 | 30.2* | 51.7* | - | - |
| DocOwl 1.5 (Hu et al., 2024a) | 1344 | 597 | 81.6 | 70.5 | 50.4 | 49.3 | 68.8 | 39.8 | 80.4 | 48.3 | 51.8 |
| Llava Next 34B (Liu et al., 2024b) | 672 | 573* | 78.2 | 67.3 | 45.1* | 70.3 | 69.5 | 47.5* | 68.9* | 43.2* | 46.5* |
| GPT4V (OpenAI, 2023) | - | 645 | **88.4** | 78.5 | 75.1 | 78.2 | **78.0** | 45.5* | 69.3* | 48.9* | 41.2* |
| Gemini Pro (DeepMind, 2023) | - | **659** | 88.1 | 74.1 | **75.2** | 73.9 | 74.6 | 32.3* | 67.9* | 38.7* | 34.6* |
| Xcomposer2 (Dong et al., 2024) | 490 | 511 | 59.6 | 72.7 | 32.9 | 78.7 | 66.1 | 28.7 | 62.3 | 34.2 | 49.3 |
| TextSquare (ours) | 700 | 622 | 84.3 | **79.4** | 51.5 | 79.0 | 66.8 | 49.7 | **84.2** | 53.2 | 71.8 |

**Document-Oriented Benchmark.** While the documents have a clean background, dense text and complex typography pose distinct challenges. To effectively evaluate our model, we select

Table 2: Quantitative comparison of our model with existing MLLMs on representative General VQA and hallucination evaluation benchmarks. VizWiz and POPE are relevant to both VQA and hallucination. Following Cogagent, we evaluate the adversarial part of POPE.

| Method | General VQA and Hallucination Evaluation | | | | |
|---|---|---|---|---|---|
| | VizWiz | VQAv2 | GQA | POPE$^{adv}$ | Average |
| Qwen-VL (Bai et al., 2023) | 35.2 | 79.5 | 59.3 | - | - |
| Monkey (Li et al., 2023c) | 61.2 | 80.3 | 60.7 | 80.3* | 70.6 |
| Cogagent (Hong et al., 2023) | 36.7* | **83.7** | 62.3* | 85.9 | 67.2 |
| DocOwl 1.5 (Hu et al., 2024a) | 43.5* | 68.0* | 48.5* | 79.7* | 59.9 |
| Llava Next 34B (Liu et al., 2024b) | 63.8 | **83.7** | **67.1** | 83.4 | 74.5 |
| GPT4V (OpenAI, 2023) | 64.9* | 77.2 | 48.4* | 79.6* | 67.5 |
| Gemini Pro (DeepMind, 2023) | 42.8* | 71.2 | 52.2* | 84.5* | 62.7 |
| Xcomposer2 (Dong et al., 2024) | 58.9* | 81.8 | 64.5 | 78.5 | 70.9 |
| TextSquare (ours) | **71.4** | 78.0 | 64.5 | **86.6** | **75.1** |

representative benchmarks, including DocVQA (Mathew et al., 2021), ChartQA (Masry et al., 2022), and InfographicVQA (Mathew et al., 2022). The results, detailed in Table 1, show that TextSquare outperforms all the open-source models in these three document-oriented VQA tasks with an average improvement of 3.5%, specifically, DocVQA 84.3% *vs.* 81.6% (Cogagent and mPLUG-DocOwl 1.5), ChartQA 79.4% *vs.* 72.7% (Intern-Xcomposer2), InfographicVQA 51.5% *vs.* 50.4% (mPLUG-DocOwl 1.5). On the ChartQA dataset, TextSquare outperforms GPT4V and Gemini Pro by a slight margin. Note that TextSquare employs an image resolution of 700, which is smaller than most document-oriented MLLMs. Our model relies on comprehensively high-quality VQA information specific to the text in the document, improving its ability to recognize and understand various document elements such as text, diagrams, infographics, and so on. If the image resolution is further increased, it is believed that the model performance will be further improved, as demonstrated by Monkey et al.

**Scene Text-centric Benchmark.** The ability to answer text-based questions in images becomes an important aspect of the answering task, as textual information is usually present in real-world scenes. In the evaluation, we utilize two datasets: TextVQA (Singh et al., 2019) and AI2D (Kembhavi et al., 2016). As shown in Table 1, in this scenario, although TextSquare achieves SOTA performance on the AI2D dataset, there is no major improvement over our baseline Intern-Xcomposer2, which might be since Intern-Xcomposer2 has been adequately optimized with high-quality in-domain data.

**Table VQA Benchmark.** Due to the complex structure of tables and the dense text, understanding the content of tables remains a challenging issue. To evaluate the performance of the comprehension of table content and structure, we choose two widely utilized datasets, Wiki Table Questions (WTQ) (Pasupat & Liang, 2015) and Table Fact (TabFact) (Chen et al., 2019), as shown in Table 1. On the Table VQA benchmarks, TextSquare achieves optimal performance among the leading models with an average 3.0% improvement. This demonstrates that our model has reached a new level of table understanding, where high-quality generated table VQA and reasoning data play a key role.

**Text-centric KIE Benchmark.** Text-centric key information extraction tasks are frequently encountered in the information processing of various types of products, certificates, and receipts. We select a receipt information extraction dataset (SROIE) (Huang et al., 2019) and a product information extraction dataset (POIE) (Kuang et al., 2023), and the KIE task is converted to the VQA task. TextSquare achieves optimal performance in both datasets, with a major average lift of 14.8% (shown in Table 1). It is worth noting that no training set of POIE is added to the training set, and there is not much data in the domain of product scenarios. This illustrates the extensive textual comprehension capabilities of our model.

**OCRBench.** OCRBench (Liu et al., 2023) is a comprehensive benchmark consisting of 29 OCR-related assessments, with text recognition, formula recognition, text-centric VQA, KIE, etc. TextSquare achieves optimal performance in OCRBench except for the closed-source models and becomes the first MLLM that exceeds 620 points with about 10B parameters. It indicates that the model performs well in both text-centric perception and comprehension tasks, especially in text recognition, where little in-domain data is included in the training set.

**General VQA and Hallucination Evaluation Benchmark.** General VQA requires learning visual and textual information and a deep understanding of their inter-relationships. For general VQA, we validate on four benchmarks: VizWiz (Gurari et al., 2018), VQAv2 (Goyal et al., 2017), GQA (Hudson & Manning, 2019), and POPE (Li et al., 2023b). The VizWiz and POPE benchmarks are also relevant for hallucination evaluation. The results are shown in Table 2. On VQAv2 and GQA, TextSquare does not have a significant degradation compared to InternLM-Xcomposer2 and still maintains comparable performance. TextSquare exhibits superior capabilities in VizWiz and POPE, outperforming the closest competing method by an average of 3.6%. These results highlight our approach's effectiveness, which can also mitigate model hallucinations, particularly with large-scale instruction tuning. We observe that it is partly attributed to the high-quality reasoning data that provides detailed explanations for VQA.

## 5.3 QUALITATIVE ANALYSIS

As illustrated in Figure 4, TextSquare has a formidable capability to provide plausible explanations of the answers to questions in various text-centric VQA scenarios. Figure 4(a) shows that TextSquare can understand and process numerical data within the text, enabling it to answer questions that require basic mathematical reasoning. Figure 4(b) shows the ability to understand textual content and provide approximate location in dense text. Figure 4(c) shows the comprehension of table structure and the ability to extract contextual information relevant to the question.

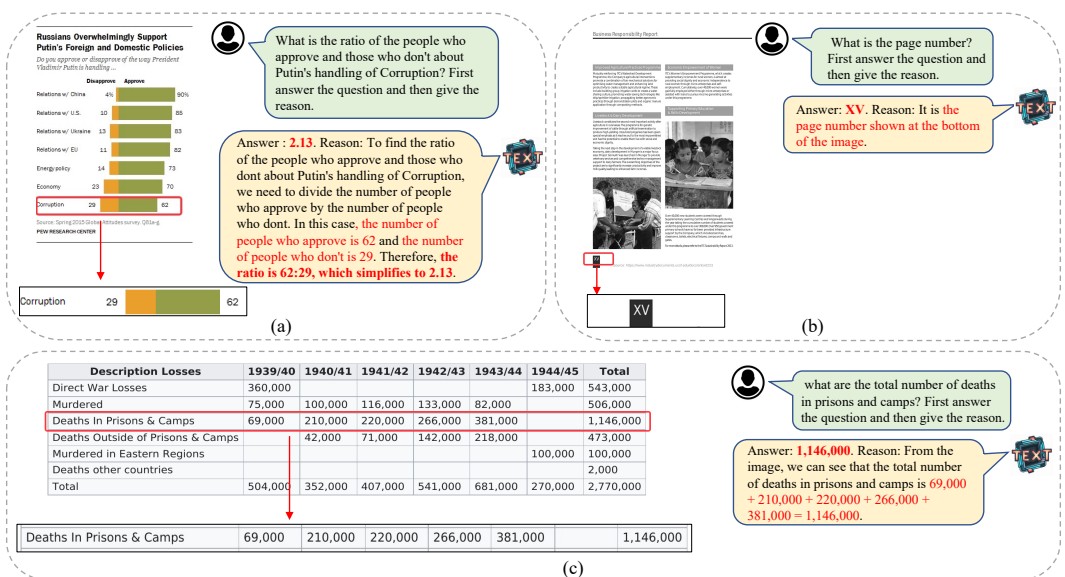

Figure 4: Qualitative results of VQA and reasoning for various text-centric scenarios.

Table 3: Ablation study on Incorporating Square-10M for Instruction Tuning.

| Model | OCRBench | DocVQA | ChartQA | InfoVQA | WTQ | SROIE | Average |
|---|---|---|---|---|---|---|---|
| Xcomposer2* | 571 | 74.8 | 73.2 | 41.6 | 40.3 | 44.7 | 54.9 |
| TextSquare | 622 | 84.3 | 79.4 | 46.2 | 49.7 | 53.2 | 62.6 |

## 5.4 ABLATION STUDY

**The Effect of Incorporating Square-10M for Instruction Tuning.** To verify the effectiveness of Square-10M, we fine-tune the baseline model InternLM-Xcomposer2 on the public text-centric VQA instruction tuning dataset (consistent with Monkey's training data). As shown in Table, TextSquare substantially outperforms Xcomposer2* (fine-tuned) on various text-centric VQA benchmarks by 7.7%, which corroborates that Square-10M can fully exploit MLLM's ability in text-centric VQA scenarios and that a massive, high-quality instruction tuning data has a major performance improvement.

Table 4: Ablation study on the evaluation step in the Square strategy.

| Evaluation | DocVQA | ChartQA | WTQ |
|---|---|---|---|
| w/ | 84.3 | 79.4 | 49.7 |
| w/o | 81.7 | 77.2 | 46.9 |

Table 5: Ablation study on the VQA Reasoning data of Square-10M.

| Reasoning Data | DocVQA | ChartQA | $POPE^{adv}$ | WizViz |
|---|---|---|---|---|
| w/ | 84.3 | 79.4 | 86.5 | 71.4 |
| w/o | 82.9 | 78.1 | 83.8 | 68.2 |

Table 6: Ablation study of the image categories of Square-10M.

| | DocVQA | InfoVQA | TabFact | WTQ |
|---|---|---|---|---|
| With all data | 84.3 | 51.5 | 84.2 | 49.7 |
| Without Tables | 84.1 | 50.9 | 68.7 | 35.9 |
| Only with Tables | 61.2 | 38.5 | 85.4 | 51.7 |
| Without Documents | 64.7 | 42.2 | 82.0 | 46.4 |
| Only with Documents | 83.9 | 52.6 | 63.3 | 33.5 |

**The Effect of Evaluation Step of the Square Strategy.** As shown in Table 4, there is a distinct improvement in model performance after incorporating the evaluation of the generated VQA data, which verifies that the evaluation step of the Square strategy improves the quality of VQA instruction tuning data.

**The Effect of VQA Reasoning Data on Model Performance and Hallucination Evaluation.** From Table 5, we can find that VQA Reasoning data is helpful in both improving VQA performance and mitigating hallucinations. Specifically, regarding enhancing VQA performance, there is a 1.4% and 1.3% gain on DocVQA and ChartQA. In terms of mitigating hallucinations, there is a 2.7% and 3.2% gain on POPE and WizViz.

**The Effect of Different image categories of Square-10M.** We conduct ablation studies about the categories of images (Tables and Documents) in Square-10M. As shown in Table 6, category-specific data significantly enhances the performance of the corresponding benchmark, as well as offering a slight boost to other benchmarks.

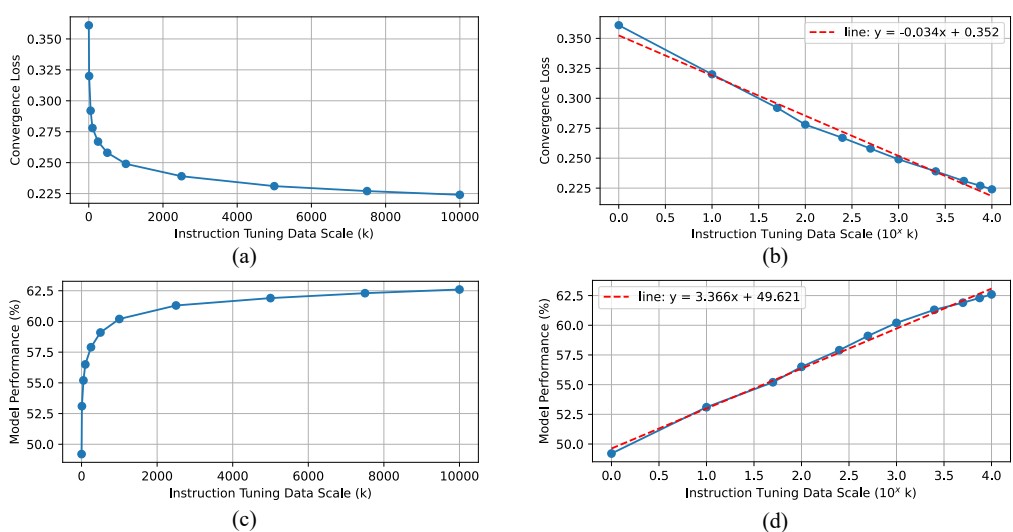

Figure 5: The relationship between instruction tuning dataset scale, convergence loss, and model performance in text-centric VQA scenarios. Figure (a) and Figure (b) show the relationship between data scale and convergence loss, distinguished by a scaling of the horizontal coordinate of Figure (b) with $\log_{10}$. Figure (c) and Figure (d) show the relationship between data scale and model performance, distinguished by a scaling of the horizontal coordinate of figure (e) with $\log_{10}$.

## 5.5 RELATIONSHIPS BETWEEN INSTRUCTION TUNING DATA SCALE, CONVERGENCE LOSS, AND MODEL PERFORMANCE

To explore the relationship between instruction tuning data scale, convergence loss, and model performance, we conduct a series of 10 experimental sets with different volumes. These experiments utilize Square-10M and specialized in-domain instruction tuning datasets. The average performance of the models is evaluated on DocVQA, ChartQA, InfoVQA, WTQ, and SROIE. As shown in Figure 5(a)(b), we observe a consistent decline in convergence loss with increasing data scale, albeit at a decelerating rate. The relationship between the convergence loss and the instruction tuning data scale approximately conforms to a logarithmic function. Similarly, Figure 5(c)(d) illustrates that model performance is enhanced with the expansion of instruction tuning data, yet the rate of improvement diminishes. This relationship also aligns with a logarithmic function. Holistically, there is a corresponding scaling law in the instruction tuning phase in text-centric VQA scenarios: model performance is directly proportional to the logarithm of the data scale. This insight is instrumental in guiding the development of larger datasets and in forecasting model performance.

## 6 LIMITATION

While yielding notable outcomes across diverse scenarios, our approach encounters certain limitations. Primarily, the processing of large-scale datasets necessitates an extensive array of GPUs for prolonged training periods. This requirement significantly escalates the overall training costs. Furthermore, despite the advancements introduced by the Square strategy in enhancing the quality of synthetic data, it falls short of achieving the nuanced accuracy and complexity characteristic of human-generated data.

## 7 CONCLUSION

This paper presents the Square strategy for constructing a high-quality text-centric instruction tuning dataset(Square-10M). Leveraging this dataset, TextSquare significantly surpasses recent open-source models and achieves performance comparable to GPT4V across various benchmarks. Furthermore, we derive the relationship between the scale of instruction tuning datasets, convergence loss, and model performance, offering insights into developing even larger datasets. Our data-centric approach reevaluates the significance of instruction-tuning data in text-centric VQA, underscoring that both the quantity and quality of data are pivotal for optimal model performance. We maintain a steadfast belief in the potential for advancing the quantity and quality of data as a means to bridge the divide between open-source models and their industry-leading counterparts.

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

Table 7: Detailed data sources of the images in Square-10M

| Chart | Chart2Text, PlotQA, FigureQA, DQA, AutoChart, DeepRuleDataset, CHART-Info |
|---|---|
| Table | FinTabNet, PubTables, WTW, TRUL, TabRecSet |
| Document | DocEdit, DUDE, FUNSD, PubLayNet, PDFVQA, CCPDF |
| Slide | PPTC, ISI-PPT, UniDoc |
| Screenshot | LightShot13k, Screen Annotation dataset, WebScreenshots, ScreenQA |
| Receipt | CORD, SROIE, WildReceipt |
| StreetView | ICDAR2013, ICDAR2015, ICDAR2017-MLT, MSRA-TD500, COCOText v2, TextOCR, Total-Text |
| WebImages | LAION-OCR, OpenImages V6 |
| E-Commerce | Amazon Product, Shopee Product |

## A   APPENDIX / SUPPLEMENTAL MATERIAL

### A.1   DATA CONSTRUCTION

Table 7 presents the detailed data sources of the images in Square-10M.

### A.2   RELATED WORK

#### A.2.1   MULTI-MODAL LARGE LANGUAGE MODELS

Recent work has increasingly focused on introducing visual knowledge into LLMs (Zhu et al., 2023; Bai et al., 2023; Dai et al., 2024). General attempts connect a visual encoder and an LLM with intermediate modules like Projector (Liu et al., 2024c), Q-Former (Li et al., 2023a), Perceiver Resampler (Alayrac et al., 2022), etc, and go through pre-training alignment and instruction fine-tuning for vision-language understanding.

Several researches (Ye et al., 2023a; Feng et al., 2023b;a; Yu et al., 2023; Wei et al., 2023; Wan et al., 2024; Luo et al., 2024; Liu et al., 2024a) propose to enhance MLLMs' capabilities in understanding textual elements (OCR, text-centric VQA, etc). Among them, mPLUG-DocOwl (Ye et al., 2023a) creates novel instruction tuning datasets to enhance the tuning process. TextMonkey (Liu et al., 2024d) adopts shifted window attention and filters out significant tokens. DocPedia (Feng et al., 2023a), and HRVDA (Liu et al., 2024a) enlarges input resolution to bridge the gap between MLLMs and visual document understanding. Despite the extraordinary progress of existing open-source MLLMs, they still suffer from the huge gap against SOTA closed-source models like GPT4V (OpenAI, 2023) and Gemini Pro (DeepMind, 2023).

#### A.2.2   TEXT-CENTRIC VISUAL QUESTION ANSWERING

Text-centric Visual Question Answering aims to understand the interactions between the image's textual and visual elements. Donut (Kim et al., 2022) first proposes an end-to-end training method based on a Transformer without OCR. Pix2Struct (Lee et al., 2023) introduces a variable-resolution input representation to adapt to document images. DoCo (Li et al., 2024) enhances the visual representation of the image encoder in MLLMs by aligning the document object of multi-modal inputs. BLIVA (Hu et al., 2024b) enlarges the input token space by concatenating learned query embeddings and encoded patch embeddings. Several studies (Feng et al., 2023b; Wang et al., 2023; Zhang et al., 2023) have performed data-centric attempts in this regard. UniDoc (Feng et al., 2023b) construct 600k document-oriented image-text pairs from PowerPoint presentations. LLaVAR (Zhang et al., 2023) and TG-Doc (Wang et al., 2023) prompt text-only GPT-4 to generate conversations for text-rich images by integrating OCR results into the instructions. These researches are restricted to small-scale annotations or generation based on uni-modal inputs.

#### A.2.3   GENERATING INSTRUCTION-TUNING DATA VIA LLMS

The success of LLMs has inspired recent work to employ them as training data generators (Chen et al., 2023; 2024; Wang et al., 2022; Shao et al., 2023). In this regard, we anchor on generating

instruction-tuning data. Self-Instruct (Wang et al., 2022) took the initial step towards synthesizing instructions via language models and improving the instruction-following capabilities. Llama-GPT4 (Peng et al., 2023) uses GPT-4 to generate instruction-tuning data for LLM fine-tuning. Synthetic Prompting (Shao et al., 2023) leverages a few handcrafted examples to prompt LLMs to generate more examples. Bonito (Nayak et al., 2024) converts unannotated text into task-specific training datasets for instruction tuning. Recently, ALLAVA (Chen et al., 2024) employs GPT4V to generate reasoning instructions and detailed answers from unlabeled images. All of the above attempts suffer from the low quality of the generated data and are typically performed on a small scale. In contrast, we collect millions of text-rich images and devise comprehensive generating methods and filtering rules to ensure the quality of the instruction tuning dataset.

## A.3 EXPERIMENTS

### A.3.1 SUMMARY OF THE EVALUATION BENCHMARKS

We summarize the evaluation benchmarks used in this paper in Table 8.

Table 8: Summary of the evaluation benchmarks.

| Benchmark | Description | Split | Metric |
|---|---|---|---|
| DocVQA | VQA on document images | test | ANLS |
| ChartQA | VQA on charts with visual and logical reasoning | test | Relaxed Accuracy |
| InfoVQA | VQA on infographic images | test | ANLS |
| AI2D | Multiple choice VQA on science diagrams | test | Accuracy |
| TextVQA | VQA involving reading and reasoning about text | val | VQA Score |
| WTQ | VQA on semi-structured HTML tables sourced from Wikipedia | test | Accuracy |
| TabFact | 'Yes' or 'No' choice VQA about tables | test | Accuracy |
| SROIE | Key information extraction from receipts | test | Accuracy |
| POIE | Key information extraction on product images | test | Accuracy |
| VizWiz | Answering visual questions from blind people | val | VQA Score |
| VQAV2 | Open-ended VQA about natural images | val | VQA Score |
| GQA | Real-world visual reasoning and compositional question answering | test-dev | Accuracy |
| POPE | Yes-or-No VQA to assess the object hallucination problem | test(adversarial) | F1 Score |
| MTVQA | Multilingual text VQA includes 9 languages and diverse scenarios | test | Accuracy |

### A.3.2 ZERO-SHOT PERFORMANCE ON MULTILINGUAL TEXT-CENTRIC VQA

To ascertain the impact of the Square-10M dataset on the generalizability of TextSquare, we undertake a zero-shot test in multilingual text-centric VQA scenarios. MTVQA (Tang et al., 2024) is a comprehensive benchmark to evaluate the model performance on multilingual visual text understanding, including nine languages and diverse text-rich scenarios. As illustrated in Table 10, TextSquare's performance across nine languages outperforms the state-of-the-art open-source models. This outcome affirms our model's robustness and our approach's efficacy.

### A.3.3 PERFORMANCE ON GENERAL MULTI-MODAL UNDERSTANDING BENCHMARKS

We evaluate TextSquare on general multi-modal understanding benchmarks. As shown in Table 9, there is a slight performance drop and TextSquare still outperforms Gemini-Pro, which is acceptable.

Table 9: Performance on general multi-modal understanding benchmarks.

| | MMB | MME |
|---|---|---|
| Xcomposer2 | 79.6 | 2242.7 |
| Gemini-Pro | 73.6 | 1933.3 |
| TextSquare | 76.8 | 2068.5 |

Table 10: Zero-shot performance of open-source MLLMs on the MTVQA (Tang et al., 2024) benchmark. The best results of each language are bolded.

| | AR | DE | FR | IT | JA | KO | RU | TH | VI | Average |
|---|---|---|---|---|---|---|---|---|---|---|
| *Open-Source MLLMs* | | | | | | | | | | |
| DeepSeek-VL (Lu et al., 2024) | 0.6 | 14.2 | 15.3 | 15.2 | 2.9 | 3.8 | 1.6 | 0.9 | 5.2 | 6.6 |
| YI-VL-34B (AI et al., 2024) | 1.7 | 13.5 | 15.7 | 12.1 | 4.8 | 5.2 | 0.8 | 3.5 | 4.1 | 6.8 |
| Llava-Next-34B (Liu et al., 2024c) | 3.3 | 24.0 | 28.0 | 22.3 | **3.6** | 6.1 | 2.6 | 0.4 | 9.8 | 11.1 |
| TextMonkey (Liu et al., 2024d) | 2.0 | 18.1 | 19.9 | 22.1 | 4.6 | **7.2** | 3.2 | 0.9 | 11.1 | 9.9 |
| mPLUG-DocOwl 1.5 (Ye et al., 2023a) | 1.0 | 13.9 | 14.9 | 18.2 | 2.9 | 5.0 | 2.0 | 0.9 | 6.4 | 7.2 |
| TextSquare | **3.7** | **27.0** | **30.8** | **26.7** | 3.2 | **7.2** | **6.7** | **5.2** | **12.4** | **13.6** |

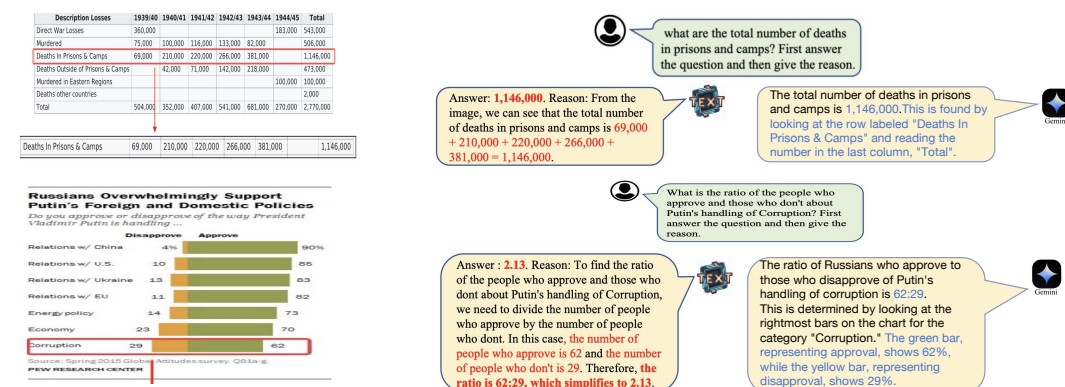

Figure 6: Visualisation of model performance comparison between TextSquare and Gemini