# OpenReview forum: "TextSquare: Scaling up Text-Centric Visual Instruction Tuning"
_ICLR.cc/2025/Conference — ICLR 2025 Conference Withdrawn Submission_

### Official Review · Reviewer_B15F · 2024-10-31

**Soundness:** 3
**Presentation:** 3
**Contribution:** 2
**Rating:** 5
**Confidence:** 4

**Summary:**

This paper builds a large-scale instruction-tuning dataset for text-centric VQA by distilling proprietary models (GPT-4V, Gemini). First, 3.8M text-rich images are manually collected, containing charts, documents, slides, tables, screenshots, street views, receipts, etc. Then, proprietary MLLMs are used to generate question-answer pairs for each image, and to provide a rationale for the answers. The MLLMs are further used to assess the quality of the annotations with self-evaluation and consistency methods and to filter the data. After filtering, 9.8M question-answer pairs are kept. The generated Square-10M dataset is paired with in-domain datasets and used to finetune an open-source MLLM: InternLM-XComposer-2. The finetuned MLLM, TextSquare, is evaluated on a suite of text-centric benchmarks, outperforming existing open-source MLLMs while being competitive (and sometimes slightly outperforming) proprietary MLLMs. In addition, TextSquare outperforms existing methods on (discriminative) hallucination benchmarks.

**Strengths:**

* Improving the OCR capabilities of MLLMs is a relevant research direction given the current real-world applications of such models.
* The paper is clear and easy to follow.
* The efficacy of the dataset in finetuning MLLMs for text-centric VQA is thoroughly evaluated on a comprehensive suite of benchmarks.
* The paper includes a comprehensive set of ablations on different subsets of the collected/generated data.

**Weaknesses:**

* The novelty of the paper is limited. For instance, previous works have also distilled proprietary MLLMs into large-scale datasets to finetune open-source models (e.g. [1]).
* Proprietary models are used to generate synthetic data to finetune open-source models. This means the performance of finetuned models is upper bounded by that of proprietary models. In cases where TextSquare's performance surpasses that of proprietary models might be just due to better prompting.
* Eliciting a rationale after providing the answer (figure 4) is not helpful to improve answer quality due to left-to-right generation.
* The design choices for the model architecture and finetuning recipe are not justified or ablated.
* It is expected that a model trained with 10M more datapoints performs better, so it is not surprising that TextSquare outperforms InternLM-XComposer-2.

[1] Liu, Haotian, et al. "Visual instruction tuning." Advances in neural information processing systems 36 (2024).

**Questions:**

* How can TextSquare achieve better performance than proprietary models if it was trained with synthetic data generated by them?
* In line 268, what does the initial 82.6% accuracy value correspond to?
* Missing relevant DocVQA datasets such as Docmatix [1].
* Can authors provide any insights on why finetuning on text-centric VQA data helps reducing hallucinations?
* Why does TextSquare's accuracy on InfoVQA differ between tables 1 and 3?

[1] Laurençon, Hugo, et al. "Building and better understanding vision-language models: insights and future directions." arXiv preprint arXiv:2408.12637 (2024).

---

### Official Review · Reviewer_TK4o · 2024-11-03

**Soundness:** 3
**Presentation:** 2
**Contribution:** 2
**Rating:** 3
**Confidence:** 4

**Summary:**

Recent research in multimodal large language models has greatly advanced text-centered VQA, with closed-source models such as GPT4V and Gemini outperforming open-source models. Challenges include insufficient instruction-adjusted data and domain inconsistencies. This paper describes the Square strategy for creating large, high-quality datasets (Square-10M) for instruction adjustment.Square includes the steps of self-questioning, answering, reasoning, and evaluating, which improves data quality and reduces illusions. TextSquare based on Square-10M outperforms existing models and demonstrates that inference data improves VQA performance. Experiments show that extending instruction tuning data reduces convergence loss and improves model performance.

**Strengths:**

1) This work constructs a high-quality dataset, Square-10M, performs full open-source data collection, and generates it using innovative build links;
2) The dataset makes current open source models better on a variety of benchmarks, some of which are comparable to closed-source multimodal macromodels
3) The correlation between data size, loss of convergence, and model performance for text-centered VQA instruction tuning is demonstrated through thorough experiments

**Weaknesses:**

1) The construction of the dataset in this work relied too much on the Gemini model and did not demonstrate the effectiveness of this construction logic on other models;
2) The construction step of the dataset, the Square strategy, did not strike the reviewers as novel, preferring it to be a variant step similar to Chain of Thoughts;
3) The magnitude of the dataset is relatively large, and the balance between the overhead of the construction costs and the benefits derived is open to discussion.

**Questions:**

1) Provide a link to a demo of this dataset, or partially open source it, and add the impact of different magnitudes of the dataset on the performance of the model;
2) Add experiments comparing other dataset construction ideas to demonstrate the uniqueness and necessity of the Square method with examples;
3) Demonstrate the same stability and superiority of the construction method on other callable commercial MLLMs (besides Gemini). Propose experiments with smaller subsets of the data to demonstrate the necessity of the full dataset size.
4) Layout of the paper to be adjusted

---

### Official Review · Reviewer_oAWF · 2024-11-03

**Soundness:** 3
**Presentation:** 2
**Contribution:** 2
**Rating:** 3
**Confidence:** 4

**Summary:**

In this paper, a prompting strategy for generating text-centric visual instruction-tuning data is proposed. The authors used a commercial VLM (Gemini) to generate a large amount of data, which was subsequently used to fine-tune an open-source VLM, achieving considerably strong performance across a wide range of text-centric VLM benchmarks.

**Strengths:**

- The model performance is strong. The paper presents a feasible solution for achieving closed-source GPT-4V level performance on text-centric benchmarks with an open-source model.
- The paper scaled the data to 10M and demonstrated its effectiveness by visualizing the scaling trend.

**Weaknesses:**

This is a solid paper if evaluated as an engineering report. However, as an ICLR submission, it falls short in terms of novelty and scientific contributions, and the ablations are insufficient.

 - Weak Novelty: The approach is essentially self-instruction and knowledge distillation. The proposed prompting methodology (Square) appears to be a straightforward implementation and lacks ablations to prove the effectiveness of each step.
 - Limited Scientific Findings: Although the abstract lists some “findings,” they primarily summarize strong performance, and it's somehow expected that adding chain-of-thought data could improve performance, and the log-linear relationship between model performance and training data volume can be observed -- they have already been demonstrated in NLP. It’s not particularly surprising to observe this in text-centric VQA.
 - Use of Gemini: The comparison between different VLMs in section 3.5 is unclear and seems insufficient.
    - It’s unclear what “10 questionnaires” means. Does it imply that 10 participants labeled the 4x1k QA sets generated by the 4 LLMs?
    - The definition of “meaningful” is not clearly stated in the questionaire.
    - In “considering the time cost, price, and quality of the data generated, Gemini-pro is our best choice…,” details on the time cost for each VLM and their respective prices should be included. Clarification on how the trade-off was made is needed.
 - API Cost and Dataset Availability: The paper used a commercial VLM (Gemini) to generate a substantial amount of data, but it does not detail the API cost, which is likely to be significant. Additionally, there are no plans to release the constructed dataset, limiting its contribution to the community.
- Typo:
    - Line 037: visual question-answering<missing space>(VQA)
    - Line 103: training convergence loss? What is it?

**Questions:**

N/A

---

### Official Review · Reviewer_D9VE · 2024-11-04

**Soundness:** 2
**Presentation:** 3
**Contribution:** 2
**Rating:** 3
**Confidence:** 4

**Summary:**

The paper proposes Square-10M, a large-scale text-centric visual instruction tuning dataset constructed using closed-source MLLMs, and demonstrates its effectiveness through TextSquare, an 8.6B parameter model achieving competitive performance on various benchmarks.

**Strengths:**

Square-10M represents a significant scale for text-centric VQA instruction tuning.

The data collection strategy provides a clear pipeline for generating high-quality instruction tuning data through self-questioning, answering, reasoning, and evaluation.

The paper offers valuable insights into scaling laws for text-centric VQA instruction tuning, showing logarithmic relationships between data scale and model performance.

**Weaknesses:**

Limited Technical Novelty:

The core methodology primarily combines existing techniques and reliance on closed-source model. The main finding that more high-quality instruction data improves performance is expected.

There are lack of comparison with automatically generated OCR datasets such as webpages screenshots and pdf parsing library.

The evaluation benchmark. The author-reported numbers come from the December 2023 Gemini technical report. However, GPT-4V and Gemini Pro have gone through multiple iterations, with more recent performance numbers reported by Molmo (Deitke et al. 2024) showing significantly different results:

DocVQA: Molmo reports 93.1% for Gemini vs paper's 88.1%
AI2D: Molmo reports GPT-4V at 89.4% and Gemini Pro at 94.4%, compared to the paper's 78.2% and 73.9% respectively

**Questions:**

This has several important implications:

Evaluation:

The evaluation section needs updating to reflect current state-of-the-art performance. The paper's claims about beating closed-source models need to be reassessed, given the updated numbers
GPT-4o results are notably absent from the comparison.


Interesting Methodological Insight:

Given the timeline, the authors seem to have newer version of Gemini for data generation while comparing against older benchmark numbers creates an interesting dynamic
This might explain how TextSquare achieved superior results - it's effectively being trained on distilled knowledge from a more advanced version of Gemini than the one it's being compared against. One possibility is that Gemini had also went through data iteration and improved its OCR heavy data.
This observation raises important questions about the role of model iteration and knowledge distillation in advancing open-source models. I think paper also should hit the snapshot version of the model and explicitly mention their version to enhance reproducibility.

---

### Note · Authors · 2025-01-21

I have read and agree with the venue's withdrawal policy on behalf of myself and my co-authors.